# A Trinitarian Ascent: How Augustine's Sermons on the Psalms of Ascent Transform the Ascent Tradition

Mark J. Boone

Department of Religion and Philosophy, Hong Kong Baptist University, Hong Kong, China;
markjboone@hkbu.edu.hk or platoandaugustine@gmail.com

**Abstract:** Augustine's sermons on the Psalms of Ascent, part of the *Enarrationes in Psalmos*, are a unique entry in the venerable tradition of those writings that aim to help us ascend to a higher reality. These sermons transform the ascent genre by giving, in the place of the Platonic account of ascent, a Christian ascent narrative with a Trinitarian structure. Not just the individual ascends, but the community that is the church, the body of Christ, also ascends. The ascent is up to God, the *Idipsum* or the Selfsame, the ultimate reality, confessed by the church as God the Father, God the Son, and God the Holy Spirit. Through the grace of the Incarnation, God the Son enables us to ascend, making himself the way of ascent from the humility we must imitate at the beginning of the ascent all the way up to Heaven, where he retains his identity as *Idipsum*. Meanwhile, the Holy Spirit works in the ascending church to convert our hearts to the love of God and neighbor. I review the Platonic ascent tradition in Plato's *Republic* and Plotinus' *Enneads*; overview ascent in some of Augustine's earlier writings; introduce the narrative setting of the sermons on the Psalms of Ascent; and analyze the Trinitarian structure of their ascent narrative. I close with some reflections on the difference between a preached Trinitarianism that encourages ascent and a more academic effort to understand God such as we find in Augustine's *de Trinitate*.

**Keywords:** Augustine; Trinitarianism; ascent; Psalms of Ascent; Platonism; *Expositions on the Psalms*; *Enarrationes in Psalmos*





## 1. Introduction

Accounts of a spiritual climb up to a transcendent reality were given long before Augustine, by Plato for example, and long after him by the likes of Boethius and Anselm.[1] (I introduce the idea of ascent briefly in Boone 2020, pp. 17, 189–90 and Boone 2023, p. 135). In what follows, I explain how Augustine's remarkable series of sermons on the Psalms of Ascent is a transformative entry in the venerable history of spiritual ascent literature in which Augustine's Trinitarian theology transforms the account of ascent as defined by Platonic philosophy. A difference in ontology makes for a difference in the account of ascent, and Augustine's Trinitarian view of the nature of ultimate reality is the foundation of his particular account. He confesses that the object of ascent is the Trinity; he explains that the Incarnation of the Son makes ascent possible; and he explains that the ascent closely involves the Holy Spirit's work in the church converting our desires to a wholehearted love of God and of one another in God. The sermons' unique ascent narrative demonstrates the power of Augustine's Trinitarian theology both to revolutionize other systems of spiritual teaching and to renew the daily life of the church. His Trinitarianism teaches us how we can ascend together towards wholeness, blessedness, and peace.

In what follows, I first briefly introduce spiritual ascent in Platonic philosophy. Then, I overview some earlier ascent writings of Augustine. Next, I introduce the ascent narrative of these sermons on the Psalms of Ascent before exploring in detail its Trinitarian structure. Finally, I propose that these sermons are in harmony with *de Trinitate*; that they do not clearly develop the same sort of analogies for the Trinity as *Trin.*; and that this is just what we would expect in a sermon to those at different stages of the ascent.

## 2. Platonic Ascent

To understand how these sermons transform the ascent genre, we must understand ascent in the philosophical tradition which most influenced Augustine. Platonic ascent writings focus on the idea of a transcendent non-physical world knowable through contemplation. The account of the non-physical world focuses on the Forms and on the ultimate reality, called by Plato "Beauty" in the *Symposium* and "the Good" in the *Republic* and called "the One" by Plotinus. Plotinus elaborates on the nature of the higher realm, positing a divine Soul and a divine Intellect at lower levels than the One.

Two texts are especially relevant to our study of Augustine: Latin translations of Plotinus' writings were, unlike Plato's *Republic*, known to Augustine directly, but he also makes good use of the powerful and influential ascent motifs of the *Republic*.[2]

### 2.1. Ascent in the Platonic Dialogues

There are several ascent entries in the Platonic corpus. The *Symposium* guides readers through an ascending series of perspectives on love (*Eros*), culminating in Socrates' speech on how he learned from the wise woman Diotima about ascent to ultimate non-physical Beauty. The *Symposium* then returns to ordinary life in a tragicomic descent depicting Socrates' failure to convert Alcibiades to the love of wisdom and the subsequent corruption of Alcibiades' character. Plato's *Phaedo* culminates in a mythical account of the universe in which good souls go on to pleasant future embodied lives, or rise up out of the murky regions of human existence altogether to enjoy a higher realm. The Myth of Er in Book X of the *Republic* gives a similar account.

These are less important than the immensely influential middle books of the *Republic*, the classic source of the guiding imagery of ascent, especially the image of the blinding sun. The first sentence of the *Republic* describes Socrates going up to Athens with Glaucon, the philosopher leading his student up towards wisdom. Sandwiched between this everyday image of ascent and the climactic Myth of Er, with its reminder to be virtuous and always to pursue wisdom for the sake of both this life and the next, we find the middle books of the *Republic*.

The account of ascent here begins with the Forms, explained in Book V. These are the universal perfect essences of things, the paradigms which are the source of whatever lesser perfections we may find in this world. There are good tables in this world, but the Form of Table is the perfection of all tableness. There are good books in this world, but the Form of Book is the perfection of all bookness. Building on the account of the Forms are three images of ascent in Books VI and VII: the Divided Line, the Sun, and the Cave. A recommended introduction to these images and their significance is Robert Wood's "Plato's Line Revisited" (Wood 1991, pp. 525–47). Wood explains how the Line in particular is a pedagogical device, itself a training exercise in the ascent. The Divided Line is a map of reality divided into two regions: the physical world or the world of Becoming, which is known by the senses, and the immaterial world or the world of Being, which is known by the mind. Further dividing both regions into lower and higher sub-regions, the account in the *Republic* introduces us to the distinction between, in the physical world, objects and their images and, in the immaterial world, the Forms and their lesser images—the truths of mathematics. We achieve true knowledge by learning to know the immaterial world, climbing up through the knowledge of mathematics to the contemplation of the Forms and, ultimately, to a vision of the chief Form, the Good, the ultimate reality and the cause of all goodness and indeed all *things*. Its image is the sun. Just as the sun in this world is the greatest good and gives life to all things, so the Good is the greatest good of the immaterial world and the origin of all.

After this map of metaphysical geography, Socrates gives us the moving Cave Analogy at the beginning of Book VII. The famous story of the Cave is multifaceted, but its main point is to further describe the ascent called for by the Divided Line and the Sun. Prisoners in the Cave represent souls mentally stuck in the physical world, ignorant of any higher world. The fire in the Cave represents the sun in our world. The prisoner freed from

the Cave and ascending to the real world, learning to see things outside the Cave after first learning to see their reflected images, is like the philosopher learning first to master mathematics and then to contemplate the Forms. Finally, he gazes at the sun outside the Cave, symbolizing the fulfillment of the philosophical quest and the pinnacle of ascent—the spiritual vision of the Good. Socrates concludes the discussion of the allegory by saying the following:

> The visible realm should be likened to the prison dwelling, and the light of the fire inside it to the power of the sun. And if you interpret the upward journey and the study of things above as the upward journey of the soul to the intelligible realm, you'll grasp what I hope to convey . . . In the knowable realm, the form of the good is the last thing to be seen, and it is reached only with difficulty. Once one has seen it, however, one must conclude that it is the cause of all that is correct and beautiful in anything, that it produces both light and its source in the visible realm, and that in the intelligible realm it controls and provides truth and understanding. . . (Plato 1997, 517b–c, p. 1135)

Its light blinds us until our eyes are strengthened through a habit of looking upward—through the moral and intellectual training that accustoms us to a higher reality. William Altman very reasonably treats the Cave Analogy as the culmination of the account of ascent in the Platonic dialogues (Altman 2018). The Good is a thing of infinite worth, exceeding for any philosophical climber even the value of one's own happiness. (But perhaps we could quibble with Altman on the grounds that knowing the Good and experiencing the greatest possible happiness are one and the same). Book VII of the *Republic* presents liberal arts training as a place to begin the quest—a pathway for exploring the lower regions of non-physical reality and working our way up to seeing the Good.

### 2.2. Ascent in Plotinus' Enneads

Plotinus is a great interpreter of Plato's writings, a great synthesizer of their various accounts and images. (See Bertozzi 2021, pp. 142–49). The highest principle in his metaphysics is the One or the Good—the ultimate reality. The next is Mind, or Intelligence or Intellect. A third divine principle is universal Soul. Plotinus introduces his own experience of knowing these divine realities through a spiritual ascent in the Eighth Tractate of the Fourth Ennead (*Enn. 4.8*; Henry/Schwyzer, Plotinus 1951). Here is the classic MacKenna translation[3]:

> Many times it has happened: lifted out of the body into myself; becoming external to all other things and self-encentred; beholding a marvelous beauty; then, more than ever, assured of community with the loftiest order; enacting the noblest life, acquiring identity with the divine; stationing within It by having attained that activity; poised above whatsoever within the Intellectual is less than the Supreme; yet, there comes the moment of descent from intellection to reasoning, and after that sojourn in the divine, I ask myself how it happens that I can now be descending, and how did the Soul ever enter into my body, the Soul which, even within the body, is the high thing it has shown itself to be. (Plotinus 1991, p. 334)

The Plotinian ascent goes beyond the body up through the levels of divinity to a direct intellectual perception of the highest reality. But Plotinus finds that his soul returns to everyday embodied life, and he wonders both why that is and why it has come to be in matter in the first place—a topic he proceeds to investigate in this passage.

Plotinus gives more details on the ascent in various other passages. In the First Tractate of the Fifth Ennead, he gives a guided tour of his metaphysics, from the human soul to the universal Soul and on up through Mind to the One (*Enn.* 5.1; Henry/Schwyzer). This account is itself an ascent, both a lesson on what we should contemplate and a rudimentary exercise in that contemplation. Souls have forgotten that they are wholly from the divine, from God (*theos*) the father (*pater*) (5.1.1). The first stage of the returning climb is learning to value our souls more than the fleeting things of this lowly world. The Fifth Ennead's

Eighth Tractate functions in much the same way, although now with a greater emphasis on Mind—on a higher level of the ascent (*Enn*. 5.8; Henry/Schwyzer). We can learn to behold the beauty (*kallos*) of Mind (*Nous*) and of that whole world of Mind (5.8.1). Here, opting not to use approximate language to convey an idea of divinity, Plotinus clarifies that the One is beyond all categories we can ascribe to things—even beyond the category of beauty (5.8.13). Meanwhile, the Fifth Ennead's Fifth Tractate focuses more on the One, which the earthly sun images—the highest level of ascent (*Enn*. 5.5; Henry/Schwyzer). The One is purely (*katharōs*) one and not one according to something else (5.5.4). In other words, it is not a unity only in relation to its own constituent parts or by comparison to something else. (MacKenna's "a unity untouched by the multiple" is appropriate; Plotinus 1991, p. 395). It is a pure simplicity, oneness itself. Besides calling it the One, it is difficult to name it (5.5.6), although we must learn to see it if we can (5.5.4). It is, however, appropriate to describe it as the Good (*Agathos*) and the cause of all things (5.5.12).

Stróżyński reminds us that ascent is above all a spiritual exercise (Stróżyński 2021, pp. 448–77). Ascent is to be accompanied by a moral life lived for the soul, not the body—a lesson borrowed from Plato's *Phaedo*. We need to become Godlike, which involves being just (*dikaios*) and holy (*hasios*) with prudence (*phronēsis*) (*Enn*. 1.2.1; Henry/Schwyzer). We must wholly enter a state of virtue (*aretē*). Plotinus discusses the virtues in detail in the rest of the First Ennead's Second Tractate, and this is not the time for a detailed explanation of his ethics. However, we should note that there are virtues that have the role of purifying the soul of undue bodily influence. (The interested reader might consult Moore n.d., sct. 4, for an overview of Plotinus' ethics; see Hadot 2004, pp. 158–60 for a more detailed look at spiritual exercises in the Neo-platonic tradition).

We must also observe one thing from Bertozzi's thorough analysis: Love (*Eros*) is how we ascend—both the cause and the method of the ascent (Bertozzi 2021). The Third Ennead's Fifth Tractate is one instructive passage on the subject. It is a commentary on Plato's *Symposium* explaining the role of Love (*Eros*) as a middle ground between embodied life and the divine, always driving the soul to climb higher.

It is now time to see what Augustine makes of this notion of spiritual ascent.

### 3. Ascent in Augustine's Earlier Writings

Plotinus' *Enneads* were known to Augustine. (A subject of much scholarship, touching on major issues in Augustine interpretation; for an introduction, see Rombs 2006, pp. 3–15). The Plotinian form of the ascent could not have failed to exert some influence on him, but so did the motifs of the *Republic*. Augustine's ascent writings are characterized by some significant continuity with the Platonic tradition and some growing differences.

The early writings present Platonic insights on the non-physicality of a higher reality; emphasize unity as a defining characteristic of divinity; teach the greater value of the soul than of the fleeting things of this world; strive to reorient human love to God; use the Platonic images of the sun and of the soul whose eye is blinded by God's light until it is strengthened; and describe a path of ascent towards God through morality as well as the Platonic priority of a liberal arts education. On the other hand, there is a confession of faith in the Trinity as the object of the ascent, which is already different from the Plotinian teaching on the gradations of divinity—One, Intellect, and Soul. There is also an appeal to the Incarnation as what makes it possible for anyone to return to God—opening up the ascent to more than just the adept philosopher. The morality taught is explicitly Christian. (For simplicity's sake we will not consider all aspects of Augustine's thought that suggest differences from Platonic philosophy. These are, however, noted by scholars—for example, Lagouanère 2012, pp. 178–79 on the use of the paradigm of the word in addition to the Platonic paradigm of intellectual vision in Augustine's epistemology).

The following is a concise introduction to some writings predating the sermons on the Psalms of Ascent, from the Cassiciacum dialogues up through the *Confessions*.

Augustine's earliest surviving writing is the first of the four Cassiciacum dialogues, *Contra Academicos* or *Against the Academics*. (One scholar who has studied ascent in the early

writings, drawing attention to Platonic themes in the Cassiciacum dialogues, is Frederick van Fleteren. For example, van Fleteren 1978, pp. 159–82). After a long dialogue analyzing the pursuit of wisdom and ultimately arguing that it can succeed, Augustine introduces the Incarnation as the final argument against skepticism. We can reach wisdom because divine wisdom has already come to us in the person of Jesus (*c. Acad.* 3.19.42; Augustinus 1970, CCL 29, p. 60, Green). His incarnation enables everyone to reach the truth, not just a few philosophically adept. Augustine's *De Beata Vita*, *On the Happy Life*, presents the pursuit of happiness—and an enhancement of the philosophical quest for wisdom—as a Christian activity (*b. Vita* 4.35–36; Augustinus 1970, CCL 29, pp. 84–85, Green). This pursuit is our return to God, fueled by faith, hope, and charity. Augustine explains that this God is one divine substance, but also a Trinity: the Supreme Measure, the Truth which is begotten of Measure and by which we know it, and a holy Admonition to return to God, who leads us to the Truth and who proceeds from the same font as the Truth. This God is light, the "secret sun" (*sol secretus*) which pours its light into our souls. We are too weak-eyed to look at it directly. Paths of ascent are laid out in the third Cassiciacum dialogue, *De Ordine* or *On Order*. With reverence for Christian authority and acknowledging his mother's piety, Augustine commends the path of following Christian authority (*Ord.* 2.5.16; Augustinus 1970, CCL 29, pp. 115–16, Green; also 2.17.45; Augustinus 1970, CCL 29, pp. 131–32, Green). This is the way we all need—the way of following Christ. This is a reconsideration of the role of morality in ascent; while it is not exactly a rejection of classical accounts of the importance of virtue, the virtues are reoriented. The emphasis of *b. Vita* is on Christian prayer and fellowship and the theological virtues of faith, hope, and love (Boone 2016, pp. 84–91). Still, for obscurer points of theology to be understood by those who can, Platonic philosophy provides helpful insight into metaphysics. So Augustine commends another path, an order of life and an order of education that would lead upward to better understanding (*Ord.* 2.8.25; Augustinus 1970, CCL 29, p. 125, Green). The soul learns to love God better by the necessary reformation of both pagan and Christian virtues. A liberal arts education is also recommended—grammar, dialectic, rhetoric, music, mathematics, geometry, and astronomy (2.9.26–2.16.44; Augustinus 1970, CCL 29, pp. 121–31, Green). This pursuit trains reason (*ratio*) to fulfill its wish to capture "the happiest contemplation of divine things" (*divinarum rerum beatissimam contemplationem*) (2.14.39; Augustinus 1970, CCL 29, p. 129, Green). For reason was desiring (*desidero* in the imperfect tense) to see a beauty (*pulchritudo*) which is single and simple (*sola et simplex*—single and pure, unmixed). Such a contemplation is not available to the bodily eyes, but to the mind that sees non-physical reality. Augustine's *Soliloquia* (*Soliloquies*) is a dialogue between Augustine and his own Reason (a word best capitalized as the name of a speaking character). Reason explains that it (*ratio*) is to God what the bodily eyes are to the sun (*Sol.* 1.6.12; Augustinus 1986, CSEL 29, pp. 19–20, Hörmann). We cannot see without healthy eyes. The things known through the liberal disciplines are like lesser things seen through the eyes; recalling the Cave Analogy, they are the lower levels of non-physical reality the knowing of which is a way up towards God. However, and less like the Cave analogy, Reason also explains that faith, hope, and charity heal the soul, strengthening the eye of reason for the vision of God. This God is described in Trinitarian terms in the prayer that begins the discussion (1.1.2–1.1.6; Augustinus 1986, CSEL 29, pp. 4–11, Hörmann). (See Boone 2016 on the account at Cassiciacum, particularly on how ascent involves love).

These early writings begin a career of writing on ascent. We will briefly observe three subsequent writings with similar emphases which also prepare Augustine to develop his mature view in the sermons on the Psalms of Ascent.

One early entry is the unfinished set of books on the liberal disciplines. Augustine's *Reconsiderations* (*Retractiones*) explains that these aimed to ascend to knowledge of the incorporeal as well as to help others ascend (*Retr.* 1.6; Augustinus n.d.: PL). This ambitious project is for those following the second path of ascent laid out in *Ord*. The method is essentially Platonic, finding in the liberal disciplines a way of climbing through corporeal to incorporeal things as if through fixed steps (*quasi passibus certis*).

Augustine writes *De Vera Religione* (*On True Religion*) to Romanianus (also the recipient of *c. Acad.*) in refutation of Manicheanism. *ver. Rel.* defends orthodox Christianity as doing a better job than Platonism at turning hearts to God. There is an extended ascent passage which supplements the salutary authority of Christian teachings with an understanding of them by reason (*ver. Rel.* 29.52; Augustinus 1962, CCL 32, p. 221, Daur). It is necessary to construct a ladder or some steps (*gradus*) going up *ad immortalia et semper manentia*—to things immortal and always remaining. Augustine overviews the ladder, leading Romanianus to reflect on the different gradations of beings (29.52–36.66; Augustinus 1962, CCL 32, pp. 220–31, Daur). Life is better than non-life, reason than mere animal life, and mind than any physical thing. Eternal truth known by the mind is higher still, as is unity. Unity as known in lesser beings leads to the unity of God: the One is the ultimate reality. This One is one with the Truth that testifies to it—God the Father and God the Son. (Observe, alongside explicit Christology, some correspondence to Plotinus in thinking of ultimate reality as One). This insight leads into an extended discussion of idolatry and the sins associated with it; of Jesus' example in refusing temptation; and of rightly ordered loves (36.66–49.97; Augustinus 1962, CCL 32, pp. 230–50, Daur). A life reorganized around these truths results in a stable happiness and in loving one's neighbor as oneself. This is the love of God poured into our hearts by the Holy Spirit, as described in Romans 5:5 (47.92; Augustinus 1962, CCL 32, pp. 247–48, Daur)—a key verse to which we will return later. (On ascent in *ver. Rel.*, especially as it pertains to love, see Boone 2020, pp. 17–23).

Augustine's *Confessions* considers spiritual ascent in a number of places, two of which I mention here. (On ascent in Book X of *Conf.*, not considered here, see Boone 2020, pp. 189–92). In a famous passage in Book VII, Augustine recounts his earlier retreat into his mind and the spiritual ascent he there undergoes, reaching up to a glimpse of God (*Conf.* 7.10.16–7.17.23; Augustinus 1990, CCL 29, pp. 103–7, Verheijen). Unable to sustain the vision, he falls back to ordinary life, and as the narrator, he explains that the only way of reliably knowing God is to follow Jesus in humility. Then, there is the famous account of the vision of God shared with his mother at Ostia (9.10.23–9.10.26; Augustinus 1990, CCL 29, pp. 147–48, Verheijen). Augustine had recently been reading Psalm 4, which mentions God the *id ipsum*, the in-itself or the selfsame (9.4.11; Augustinus 1990, CCL 29, p. 139, Verheijen). In conversation with his mother, they rise by burning affection to the *id ipsum*. There is no mention of a tragic fall back to earthly life, even as life goes on and as Augustine's mother dies days later; but there is a sweet longing for the perfect enjoyment of this goodness of God after death and after the resurrection of the dead.

## 4. The Ascent of Iduthun

Augustine inherited the ascent tradition from Platonism and had already developed it in a Christian direction in these writings. Trinitarian confession, the Incarnation of the Son of God as the way for all to reach the highest, and love as a work of the Holy Spirit are already elements of the ascent. Augustine's deepening Trinitarian theology further transforms the tradition in the sermons on the Psalms of Ascent, in which these same elements now structure the whole. Phillip Cary observes that the vision at Ostia is something like "a mutual catechetical lecture . . . where by the grace of God the words spoken and heard actually succeed in directing both minds to see with some clarity what they are trying to understand" (Cary 2008, pp. 11–12). It is not a non-repeatable mystical experience, but something which could occur "every time someone in the audience understands the significance of a good sermon" (Cary, *Outward Signs*, p. 189). The same occurs in the sermons on the Psalms of Ascent. Augustine is trying to replicate the ascent at Ostia on a wide scale, succeeding to the extent that his hearers—and readers—understand and see the same things themselves.

These sermons occur roughly some five to ten years after *Confessions* was finished around A. D. 400 or 401.[4] They are the *Enarrationes in Psalmos* 119–33—covering Psalms 119 (120 in the modern numbering, based on the Masoretic text) through 133 (134). The *Enarra-*

*tiones* are a massive set of commentaries on the Psalms, mostly composed of Augustine's sermons. (For a good introduction to the Enarrationes, see Fiedrowicz 1997 or Dupont 2020, pp. 320–37). Many of the *enarrationes* may be grouped together in sets of sermon series. The *enarrationes* on the Psalms of Ascent, themselves a particular grouping of Psalms in the Old Testament, may be considered as a series in themselves, or alternatively grouped together as part of a larger sermon series which also includes sermons on John's writings in the New Testament. These are complementary biblical commentaries with overlapping themes and a heavy emphasis on critiquing the Donatist schism. (See Ployd 2015 for a closer look).

Our priority here is the influence of Trinitarianism in this Christian pastor's maturing account of the spiritual ascent. The place to begin is the setting of the ascent narrative, which Augustine describes using several related images: Iduthun the leaper, the journey up the mountain from the valley of weeping, and the climb up to Jerusalem.

The character of Iduthun appears in Psalms 38 (39), 61 (62), and 76 (77). We may find in modern Bibles a superscription identifying the Psalm as being written to Jeduthun; these are variations of the same name following different courses in translation from Hebrew into Greek, Latin, and English (see Grove 2021, p. 86, note 2 on Jeduthun). Augustine's sermons on these three Psalms explain this mysterious figure. Iduthun is a leaper—"one who leaps across" (*En.* 38, 1; Augustine 2000; Augustinus 1956a, CCL 38, p. 401, Dekkers and Fraipont[5]). Iduthun leaps over earthly things up to heavenly ones. We should all strive to be Iduthun, but he does not simply represent an individual believer. The work of Jesus, as often considered in the *Enarrationes in Psalmos*, blends the identities of Christ, the church, and the individual believer. Iduthun represents the church on its way up to heaven. (See Grove 2021, p. 87 and McLarney 2014, p. 143). Augustine connects Iduthun to the Psalms of Ascent (*En.* 38, 2; Augustinus 1956a, CCL 38, pp. 402–3, Dekkers and Fraipont). Iduthun leaping across earthly things is the same as a climber on steps (*gradus*), a climber on a ladder (*scala*), or one flying up on wings (*pennae*). These are all the same ascent made by the love (*affectus*) of a good will (*bona voluntas*).

Augustine's series on the Psalms of Ascent begins with an introduction drawing on the Septuagint's name for these Psalms. The congregation having just heard and responded to Psalm 119 (120), he explains its label: *canticum graduum*, a song of steps or a song of ascents (*En.* 119 1; Augustine 2003; Augustinus 2001, CSEL 95/3, pp. 37–40, Gori).[6] The Septuagint term is *anabathmōn*, the genitive plural of *anabathmos*, an ascent or a set of stairs. Plotinus uses related words, for example employing a participial form of the verb *anabainō* to describe people who are climbing in the Fifth Ennead, Eighth Tractate (*Enn.* 5.8; Henry/Schwyzer). (Stróżyński 2021 considers Plotinus' use of anabainō). Augustine may not have been aware of this linguistic connection, although he was certainly aware of the connections between this idea of a climb in one text and a parallel idea in another.[7] "We too are to ascend, but we must not try to climb with our bodily feet;" Psalm 83 (84) explains that the ascent is in the heart, begins from "*the valley of weeping*", and ends in "*the place he has appointed*". Augustine frankly explains that we cannot be told what this place is, for it is beyond our present ability to comprehend. It is an *ineffabiliter locus beatitudinis*—an indescribable place of happiness. The valley of weeping means humility, referring to the Incarnation of Jesus, an example for us to follow in humbling ourselves so God may lift us up. (On the beginning of the ascent in humility, see also Ployd 2015, pp. 42–54). This same Jesus is also the mountain we are ascending.

A third image of the narrative is the church going up to Jerusalem—*not the Athens of the* Republic, *but Jerusalem!* This picture connects more directly to the original sense of the Hebrew Psalms—the Jews going up to Jerusalem to give thanks in the temple (Psalm 121 (122)). Augustine uses this as a figure of the church on its way to heaven. Jerusalem is the heavenly homeland (*patria*) of God's people (*En.* 119, 6; Augustinus 2001, CSEL 95/3, pp. 48–49, Gori). The Psalmist laments his exile because God's people sojourning on earth groan (from *gemō*) with longing for their heavenly home (119, 7; Augustinus 2001, CSEL 95/3, pp. 50–53, Gori). The heaven-bound church sings these songs. (For more on Jerusalem in these Psalms, see Ployd 2015, p. 34 and Renna 2001, p. 282).

The leaping Christian; the leaping church; the humble sinner going up to heaven by God's mercy; the climber going up by degrees, by steps, by gradations to Jerusalem; the church on its way home—these ideas all come together to constitute the ascent narrative.

With this all-too-brief introduction to the ascent in place, we will be able to look directly at its Trinitarian structure.[8] Here is one place where we still have much to learn about Augustine's Trinitarian thought. Let us look at some scholarly contributions to our understanding of Augustine's Christian theology in these sermons, and then consider what is still lacking.

Gerard McLarney's *St. Augustine's Interpretation of the Psalms of Ascent* introduces these sermons by explaining Augustine's "hermeneutic of alignment", wherein he works "to *align* or *establish continuity* between the song of the Psalmist, the Psalmist, and the lives of his readers within an overarching common framework" (McLarney 2014, p. 7). That framework is the *totus Christus*, the whole Christ, including his body the church, from Old Testament saints down through Paul and the martyrs to Augustine's congregation in North Africa centuries later, who become "coparticipants with the Psalmist and the saints who embark on the ascent, whether patriarchs, prophets, apostles, or martyrs" (McLarney 2014, p. 123). Kevin Grove's *Augustine, the Trinity, and the Church* explains how Augustine's account of memory in the ascent develops into an account of the church's work of becoming Christ by forgetting self, remembering Christ, and anticipating heaven (Grove 2021). Lewis Ayres' *Augustine and the Trinity* is an important argument for Augustine's Nicene Trinitarianism—a Trinitarianism influenced heavily by earlier patristic sources, committed to three distinct divine Persons, and neither neglecting Platonic ideas nor depending on them instead of the Bible (Ayres 2010). (See Ployd 2015, pp. 3–11 for a helpful introduction to Ayres and some related scholarship on Augustine's Trinitarianism). Adam Ployd's *Augustine, the Trinity, and the Church* connects the sermons on the Psalms of Ascent to sermons on John's writings in the New Testament, explaining the critique of Donatism in these sermons and clarifying how, as a part of that critique, Augustine's Nicene Trinitarian theology is connected to the establishment, love, and unity of the church. The Donatists were Trinitarian, but in Augustine's mind, they did not understand the importance of Christian unity as a consequence of Trinitarianism. This emphasis on unity, we may observe, is a correspondence to Plotinus as well as to Augustine's other writings (such as *ver. Rel.*, considered above). Perhaps there is some influence from Plotinus here. However, Augustine's conscious goals and influence in emphasizing the unity of the church are entirely biblical. The sources of his ideas are in particular the biblical doctrines of the Trinity, the Incarnation, and the union of Christ and the church. (See also Baker 2010, pp. 7–24).

This fine scholarly work leaves more to be done. There is some focus on ascent here or there, but little if any direct focus on the Trinitarian structure of ascent in these sermons. Similarly, there are scholars who talk about Augustine's account of ascent with an emphasis on things like the interplay of faith and reason as well as the doctrine of creation (including Cary 2000 and Harrison 2008, pp. 35–73). But there is little or no commentary on how Trinitarianism transforms the ascent genre, particularly in these sermons. We need to understand how Augustine here gives an ascent narrative in which his orthodox Trinitarianism transforms the very idea of spiritual ascent. These sermons are a Trinitarianly transformative entry in the ascent genre in which the spiritual ascent theme we have been tracing comes into its Augustinian maturity as a Trinitarian account of the church's return to God. Some aspects of ascent common to the Platonic account remain, including the idea of a non-physical reality, the importance of morality, and re-learning how rightly to value higher things. However, and as we will shortly explore in detail, in the place of the ascending philosopher is now the Christian church. The Triune God is now the object of ascent, the incarnate Christ is the beginning, and the Holy Spirit is the divine power in the ascending church. The Holy Spirit requires a reconsideration of what kind of morality to emphasize. The morality that helps us ascend is brotherly love in the church. These ideas

are key changes from the Plotinian account, although a natural development or elaboration of the account in the earlier writings.

I begin with the church's divine destination—the Trinity. I continue with the sermons' focus on the Son, who makes the ascent possible. I then consider their focus on the Holy Spirit, whose work of converting to God the desires of the ascending church is part of ascent both as its result and as the force behind its continuation.

## 5. The Church's Divine Destination

The account here concerns a Christian ascent from nadir to zenith. The object of ascent is the Trinitarian God, and those who begin the ascent are the orthodox church recognizing this doctrine. We must consider the requirement that those beginning the ascent trust in the orthodox teachings of the church; the identity of the ascending church; and Augustine's remarks on the pinnacle of ascent as the *Idipsum*, the Selfsame.

The ascent begins with trust in Trinitarian orthodoxy. In Psalm 130 (131), the Psalmist says he does not concern himself with matters too great for him; he is like a contented infant. Following variations in the Latin translations, Augustine takes the Psalmist's emphasis to be on the infant's contentedness with their mother's milk (*En.* 130, 9; Augustinus 2001, CSEL 95/3, pp. 278–80, Gori).[9] He builds on Paul's analogy of spiritual milk to explain the following: Just as solid food must be taken into the mother's flesh and converted to milk for little ones, so God the Son, the spiritual food the very angels enjoy, became enfleshed to rescue spiritually little ones. (On Jesus Christ's flesh as spiritual milk, see Ployd 2015, pp. 46–47). To be contented with mother's milk is to accept the flesh of Jesus and what we learn from it. We must not retreat *a fide lactis nostri*, literally from the faith of our milk (130, 11; Augustinus 2001, CSEL 95/3, pp. 282–83, Gori). (We must not "abandon our milky diet of faith", in Boulding's translation; Augustine 2004). This is faith in Jesus, and in the orthodoxy established by his Apostles and their successors. "Our Lord Jesus Christ is the Word of God, as John tells us . . ." (130, 9; Augustinus 2001, CSEL 95/3, pp. 278–80, Gori). "Our Lord Jesus Christ was the Word dwelling with the Father, the Word through whom all things were made". This milky faith teaches God the Father, God the Son, and God the Holy Spirit. Some heretics "wanted to argue about food beyond their capacities", not heeding the Psalm (130, 11; Augustinus 2001, CSEL 95/3, pp. 282–83, Gori). They posited divine gradations, ultimately teaching *tres dii*, three gods. They confessed that all are God, but made them out to be unequal and *non eiusdem substantiae*, not of the same substance. These heretics did not rest content with the milk provided by their mother, the church, but abandoned her. One who cannot yet see the equality of God the Father, the Son, and the Holy Spirit must "simply believe this and go on sucking" on his milk; more literally, let him trust this and take it in—*credat hoc, et sugat* (130, 13; Augustinus 2001, CSEL 95/3, p. 286, Gori). (These heretics are the Arians,[10] but Augustine's disagreement with them parallels a key difference from Plotinus—there are no gradations in God).

The church itself is ascending.[11] At one level, the church dispenses this milk, its stronger ones teaching the littler ones. At another level, the church is those very lowly ones just starting the ascent. An acceptable reading of the Psalmist's comparing himself to an infant is that we must grow past spiritual milk and move towards understanding (*En.* 130, 12–13; Augustinus 2001, CSEL 95/3, pp. 283–87, Gori). Some in the church are higher, moving past mere milk—accepting the church's proclamation of the Trinity, but also able to see and teach. Jacob saw ladders (from *scala*), and on it were people going up and down (119, 2; Augustinus 2001, CSEL 95/3, pp. 40–44, Gori). Both those ascending and those descending—*et ascendentes et descendentes*—are good folks. Some are ascending, and others are holy teachers descending to help them. (See also McLarney 2014, pp. 138–39).

In a sense, then, an individual ascends alone, beginning at the base of the ladder, drinking spiritual milk, and believing in the orthodox teaching of the Trinity. He or she later learns to know Christ himself to some extent, becoming able to take hold of "the Word who is God with God, in the form of God and equal to the Father" (*En.* 130, 11; Augustinus 2001, CSEL 95/3, pp. 282–83, Gori). In another sense, no one ascends alone. We are bound

to the life of the church. (A theme especially explored in Ployd 2015). We must accept the teachings of the church, love our neighbors, and do mercy (*fac misericordiam*); we must be unwilling to give up the peace of the church (*pacem ecclesiae noli dimittere*) like the Donatists did (130, 13; Augustinus 2001, CSEL 95/3, pp. 285–87, Gori). God taught this, which should always be in our minds, Augustine reminds us with a quote from Sirach.

The *Totus Christus* (mentioned briefly above) is the model for this—the Whole Christ. The *Totus Christus* is the idea that guides Augustine's interpretation of the Psalms. Jesus Christ is, through his incarnation and redemption of the church, eternally bound to her. (For an introduction to this idea see Dupont 2020, pp. 328–34; Fiedrowicz' account in Fiedrowicz and Müller 1996, pp. 848–55; Cameron 2012, chapter 6; or various sources cited in Boone 2023, chapter 1). She is his bride, and we cannot speak of one without speaking of the other. As Paul says, they are two in one flesh—the earthly marriage of husband and wife symbolizing the union of Christ and church. Why, then, may we not say that the two in one flesh—Christ and church—speak in one voice in Scripture (*En.* 30 2, 4; CSEL 93/1B, p. 149, Augustinus 2011; also 34 2, 1; Augustinus 1956a, CCL 38, p. 311, Dekkers and Fraipont)? This notion says a lot about the ascent. Christ is in heaven with God the Father, but his body is on earth in the person of the church. (See, for Augustine's own explanation of this, *En.* 90 2, 1; Augustinus 1956b, CCL 39, pp. 1265–66, Dekkers and Fraipont). The very body of Christ includes members in heaven (McLarney 2014, p. 163). We, even now, may be "in holy fellowship with the angelic citizens of the eternal Jerusalem" (121, 1; Augustinus 2001, CSEL 95/3, p. 84, Gori). Everyone in this community is connected in one body; "all the saints form a single person in Christ" (*unus homo in Christo*) (119, 7; Augustinus 2001, CSEL 95/3, p. 50, Gori). (See also Ployd 2015, pp. 66–73). One who can ascend and see by himself is bound to others by the cords of identity, of love, and of the responsibility to teach. This teaching also clarifies how the different branches of theology are connected. The doctrine of the Trinity is the very thing taught by the doctrine of the Incarnation, which also binds the church to Christ—Trinitarianism, Christology, and ecclesiology all blurred together.

We begin by confessing, and ascend as the church, but where are we *going*? To the *Idipsum*! Those pilgrim ascenders are climbing up to the heavenly Jerusalem, says Augustine, and that city "*shares in the Selfsame*" (*En.* 121, 5; Augustinus 2001, CSEL 95/3, pp. 90–92, Gori). This term, translated by Boulding as "the Selfsame" or as "Being-Itself" and already anticipated by *Confessions* IX, is *Idipsum* (all letters capitalized in the CSEL text—*IDIPSUM*).[12] This is the ultimate reality, the perfect being, the one who supremely exists, the one who alone can say to Moses, "I AM WHO I AM". This is God—"That which always exists unchangingly, which is not now one thing, now another . . . The eternal, for anything that is constantly changing does not truly exist . . .". We are ascending to an enjoyment of the vision of *Idipsum.* For now, the mind's eye is weak, and "this is too much to understand, too much to grasp". We must "Hold onto the flesh of Christ . . .". Jesus, as the Good Samaritan, rescues the weak and wounded sinner, bringing him for healing to the inn, which is the church. (See, on this imagery, McLarney 2014, p. 174). From there we must use our own feet and run to the city.

This is a conventional account of spiritual ascent in some respects; Plato himself dreamed of seeing the ultimate reality—the unchanging, that which supremely is. It is normal for ascent writings to say a few words about the destination before observing how the mountain peak is wrapped in clouds even as we, haltingly, take our first steps into the foothills. What is remarkable about this passage is the Trinitarian dogma using the name of God from Exodus. There where Christ thought it not robbery to be equal with God, according to Paul's letter to the Philippians—there he is *Idipsum—ibi est Idipsum* (*En.* 121, 5; Augustinus 2001, CSEL 95/3, 91, Gori). There is one *Idipsum*, but one who is *Idipsum* is also our Messiah. Perhaps this is why Augustine does not use the name of God the Father so often as he uses the name of Christ in this sermon. For the one who knows the Father is the one who knows the Son, as we read in John 14, and our responsibility as penitent sinners is to cling to Christ in order to know *Idipsum*.[13] Even the heavenly Jerusalem on the heights knows *Idipsum* by knowing Christ, for the Son is the bread even of the angels—the same

food we hope to eat in eternity in the same manner as they (130, 9–11; Augustinus 2001, CSEL 95/3, pp. 278–83, Gori).

## 6. Christ, the Way Home

We turn now to look more directly at the ascent's Christological aspects. Jesus the Christ makes the ascent possible, and indeed he himself is the way up, stretching from the valley of weeping up to the sun itself poised above the mountain whereon is set that holy city of Jerusalem (McLarney 2014, pp. 135–36). Augustine explains this in his introduction to the sermon on Psalm 119 (120). "Who is the mountain? Who else but our Lord Jesus? He made himself into a valley of weeping for you in his passion; and he is the mountain of your ascent because he remains where he has always been" (*En.* 119, 1; Augustinus 2001, CSEL 95/3, pp. 38–39, Gori). In the humility of the Incarnation, Jesus meets us, being the beginning of our ascent and the model for the humility we need. Yet he remains one with God the Father, the true God. "He is the starting point of your ascent and the goal of your ascent . . .". (See also Meconi 2013, p. 116).

At the base, we imitate his humility. At the pinnacle, we eat the bread of angels, knowing the Word of the Father through whom all creation knows God. This description of the top of the mountain in the sermon on Psalm 130 (131) accompanies the analysis of *Idipsum* in the sermon on Psalm 121 (122). That reality which exists in itself, the source of all being, is a respectable concept in the Platonic tradition. But the idea of stronger souls, having been nourished on the milk of Trinitarian dogma, feeding on Jesus who is the Word of God and one with the Father—*this* idea belongs in *a sermon*!

Other remarkable Christological remarks occur throughout this sermon series. We will look at just a few, from the sermons on Psalms 126 (127), 123 (124), 125 (126), 127 (128), 124 (125), and 132 (133).

The sermon on Psalm 126 (127) considers the line in the *Vetus Latina* translations, "*It is a waste of time for you to rise before the light*" (*En.* 126, 4; Augustinus 2001, CSEL 95/3, pp. 190–92, Gori). The original verse was probably making a point more along the lines of "Hard work won't amount to anything if God doesn't help you!" In Augustine's sermon, the verse becomes a commentary on the importance of humility before Christ with cross-references to Peter and the sons of Zebedee. We sin, as they did in the gospels, by trying to rise before the light, which means trying to rise before Christ. We can only rise *after* him. We can only ascend by following him, and this begins with imitating his humility. "Do you want to be there too, on high with him? Be humble, then, here where he too was humble".

Again reiterating the idea that Christ himself in his incarnation is at both the beginning and the end of our ascent, making that ascent possible for us, Augustine explains that *rex patriae factus est via*—the king of our homeland has been made the way thereto (*En.* 123, 2; Augustinus 2001, CSEL 95/3, pp. 119–30, Gori)! Thus we may, even while still on the way to our homeland, rejoice as people who belong there and are returning. This is our joy in hope (*in spe*), not yet in the realization (*in re*) of our hope. This is the joy of the church singing the words of the Psalm.

In the conclusion to the sermon on Psalm 125 (126), Augustine gives in more detail his interpretation of the parable of the Good Samaritan (*En.* 125, 15; Augustinus 2001, CSEL 95/3, pp. 183–84, Gori). This man had gone down from Jerusalem, an image of the fall of Adam—and of us in Adam. Because the law could not save us, both a priest and a Levite passed him by. A Samaritan stopped to help—Jesus Christ, for he, when he was in John 8 accused of being a Samaritan and of having a demon, only denied the latter. He helped us, took us to the inn, and handed us over to Paul the innkeeper, with two coins *unde curaretur*—whence would be cured that man who came down. These two coins are the love of God and of neighbor. Augustine exhorts, "If we have descended and been wounded, let us now ascend. Let us sing and make headway, so that we may arrive at our homeland".

Preparing to preach on Psalm 127 (128), and having explained the spiritual interpretation of the Psalm and the unity of Christ and the church, Augustine reminds the flock that the love described by John banishes fear (*En.* 127, 7; Augustinus 2001, CSEL 95/3,

pp. 213–14, Gori). The fear which love excludes is the fear that God may punish us with trouble on earth. There is a purer fear of God—a *castus timor*, a chaste fear. This fear is the fear of losing God because of our own sins (127, 8; Augustinus 2001, CSEL 95/3, pp. 214–17, Gori). One of these fears is like that of an adulteress who fears her husband catching her. The good, chaste fear is that of a good wife who loves her husband, fearing only his absence. Such is the love the church should have for Jesus. The church is the fruitful wife mentioned in the Psalm (127, 11; Augustinus 2001, CSEL 95/3, pp. 220–22, Gori). As Christ slept in his death, the water and blood which stand for baptism and the Eucharist flowed from his side. These are the sacraments which establish the church. Adam's sleep during which his wife was formed from his side symbolizes the same thing: the church established from Christ, and bound to him eternally. Thus the *Vetus Latina* translations refer to "*the sides of your house*". This wife bears many children, the multitudes in the church (127, 12; Augustinus 2001, CSEL 95/3, pp. 222–23, Gori). They make peace, for they grow up like olive trees, which symbolize peace (127, 13; Augustinus 2001, CSEL 95/3, p. 224, Gori).

Peace is more directly identified with Jesus at the end of the sermon on Psalm 124 (125). The inheritance (*hereditas*) and homeland (*patria*) of God's people is peace—*pax* (*En.* 124, 10; Augustinus 2001, CSEL 95/3, pp. 160–62, Gori). This very peace is Christ—*Ipsa est Christus*, whom Paul describes as the peace of the church. Peace is to be loved by the church; they love not peace who divide unity—a criticism of the Donatists. (On this theme in these sermons see Ployd 2015 and McLarney 2014, pp. 145–48, 207–11).

The love of unity is an important standard used in Augustine's critique of the Donatists. In the sermon on Psalm 132 (133), Augustine considers the unity which is good and pleasant—*bonus* and *iucundus* (*En.* 132, 7; Augustinus 2001, CSEL 95/3, pp. 328–29, Gori). It is the unity of brethren dwelling together, which the Psalmist compares to anointing oil flowing down over Aaron, who symbolizes Christ the ultimate priest. Paul describes the unity established by Christ when he says that the law of Christ is fulfilled when we bear each other's burdens (132, 9; Augustinus 2001, CSEL 95/3, pp. 331–32, Gori). The Donatists abandon this unity, and in so doing also abandon Christ (132, 6; Augustinus 2001, CSEL 95/3, pp. 326–28, Gori).

This brief look at a few Christological teachings in the sermons on the Psalms of Ascent cannot do justice to Augustine's remarkable work of thinking and rethinking the work of Christ and his intimate communion with the church while cycling through an endless supply of biblical cross-references. These sermons paint ascent with the crimson color of Christ. The whole ascent, and all its aspects as mentioned by the Psalmist, teach Christ and the church. While there are certainly elements consistent with Platonic philosophy, the ascent is the work of Christ and is taught in the Scripture. As Cameron observes of the *Enarrationes*, "The Neoplatonic spiritual ascent to God familiar from his earlier treatises is not absent, but is interrelated with a biblical spirituality rooted in Christ" (Cameron 1999, p. 292). This ascent narrative is a *preached* narrative. The Bible has superseded (although not necessarily replaced) whatever was of value in the liberal arts education Augustine had earlier considered as a path of ascent (a topic considered in Pollman 2007). Augustine's own words in the introduction to Psalm 123 (124) are a fitting summary:

> The song you have just heard being sung to you is headed, like others in this group, *A Song of Steps*. That is its title; so it is a song chanted by people mounting upwards [from *adscendo*]. Sometimes a single voice is heard and at other times there seems to be a multitude, because, though many, we are one, for Christ is one, and Christ's members are one with Christ, one in Christ. The head of all these members is in heaven. The body is toiling on earth, but it is not separated from its head, for the head looks down from heaven and cares for his body. (*En.* 123, 1; Augustinus 2001, CSEL 95/3, p. 128, Gori)

Augustine's ascent narrative is a way of telling the story of the Gospel.[14]

### 7. The Holy Spirit in the Ascent

The outcome of the Gospel is love, the work of the Holy Spirit. Although Augustine does not talk explicitly about the Holy Spirit nearly as much as about Christ in these sermons, the Holy Spirit is essential to his account. His work in the church converting our desires to the love of God and neighbor is the result of the ascent and also the energy that powers a continuing upward climb.

I first look at some illustrative passages on love and then look directly at what Augustine says about the Holy Spirit. I will focus on the work of the Holy Spirit in enabling the church rightly to love, pouring charity into our hearts to enable us to stand in God's courts, and on his organizing of Christ's church into a body, requiring an attitude of humility and love.

Psalm 121 (122) is about going up to God's house. "Love is a powerful thing", Augustine explains—*dilectio fortis res* (*En.* 121, 10; Augustinus 2001, CSEL 95/3, p. 102, Gori). Love alone meets our obligations. This Paul explains concerning love (*caritas*). Love alone if it loves (from *diligo*) with next to nothing accomplishes as much as wealthy Zacchaeus giving away half his possessions. Love is how we climb up to God's house; "We travel not on foot but by our affections [from *affectus*]" (121, 11; Augustinus 2001, CSEL 95/3, p. 103, Gori). Those who love (*diligo*) Jerusalem will have riches (or an abundance, *abundantia*) in the enjoyment of eternal life. These riches are what establish the peace of Jerusalem. Love (*dilectio* and *caritas*) is mighty, for it is aimed at God (121, 12; Augustinus 2001, CSEL 95/3, pp. 104–5, Gori). The riches of the heavenly Jerusalem consist in its participation in God the *Idipsum* and "the fullness of delights"—*plenitudo deliciarum*. That participation is by love—*caritas*. The sermon on Psalm 125 (126) builds on the power of love to aid in the ascent. Augustine explains that he preached it so that mercy (*misericordia*) may be done, for hence we are moved upward (*En.* 125, 15; Augustinus 2001, CSEL 95/3, p. 183, Gori).[15]

Augustine gives some direct remarks on the Holy Spirit who empowers this love in us. We begin by returning to the sermon on Psalm 127 (128). Having been established as the body of Christ, the church must grow in chaste love, with its proper fear (*En.* 127, 7; Augustinus 2001, CSEL 95/3, pp. 213–14, Gori). Whence comes such love? From the Holy Spirit. Jesus loves a right love (*caritas*), and loved us that we might love him back; and in order that we might be able to do so he came to us in his own Spirit—*et ut eum redamare possimus, visitavit nos Spiritu suo*. He is absent, but the church remains his bride, and he gave to her the Holy Spirit as an *arrha*, related to the Greek *arrabon* used by Paul in 2 Corinthians 5 (127, 8; Augustinus 2001, CSEL 95/3, pp. 214–17, Gori). He makes us able to love.[16]

Romans 5:5 is the key verse by which Augustine interprets the Holy Spirit's action in enabling us to love. Let us turn to the final sermon on the Psalms of Ascent—*Enarratio* 133. They will stand in the courts of God's house who stand in love (*En.* 133, 1; Augustinus 2001, CSEL 95/3, pp. 336–38, Gori). Courts (*atria*) are the more ample spaces of a house, and there is breadth (*latitudo*) in love (*caritas*). Paul explains that the love of God has been poured out or spread out (*diffusa est*) into our hearts by the Holy Spirit. So the Holy Spirit is the one who gives us the love needed to stand in God's courts. Romans 5:5 says, "*The charity of God has been poured abroad into our hearts through the Holy Spirit who has been given to us*". This pouring is just what enables us to stand (*sto*), to persevere (*persevero*), and to dwell (*habito*) in the house of God. Blessed (*beati*) are those who do so.

Finally, we return to the sermon on Psalm 130 (131). The Psalmist has not raised his eyes too high or "*walked in great matters, or in wonders above me*" (*En.* 130, 5; Augustinus 2001, CSEL 95/3, pp. 270–71, Gori). The Psalm's words remind Augustine of Simon the magician in Acts 8, who tried to buy the power of the Holy Spirit from the Apostles. There are others who want this sort of power, thinking too much of their own spiritual progress (130, 6; Augustinus 2001, CSEL 95/3, pp. 271–74, Gori). They are not humble, loving neither the peace nor the unity of the body of Christ. Paul taught us better in 1 Corinthians, teaching the unity of the church through the image of the unity of the human body. The hand, foot, eye, and ear all work together. No other part envies the eye, which sees for it, and no other part envies the ear, which hears for it. The whole body suffers when one part suffers

and works together for its healing. Thus anyone in the church may share in what Peter did on behalf of the body of Christ, and Christ spoke to Paul of being persecuted himself when Paul was persecuting the church. The words of the Psalm belong to any Christian who "honestly does what he can, not envying anyone else who can do more but rejoicing with that other member inasmuch as both of them are united in the same body . . ." (130, 7; Augustinus 2001, CSEL 95/3, pp. 274–77, Gori). The more God gives us, the humbler we ought to be. Thus Paul, having been given more abundant grace than the other Apostles, was blessed with suffering—the thorn in his flesh—to keep him humble. Learning from this, any Christian ought humbly to follow Christ and have that faith through which the heart is cleansed (130, 8; Augustinus 2001, CSEL 95/3, pp. 277–78, Gori).

The emphasis of this passage is not so much on the Holy Spirit as, once again, the church. However, in this case, the lesson concerns what the church must do in response to what the Holy Spirit has done. Paul's lesson on the unity of the body is a commentary on what the church should do with those spiritual gifts. The Holy Spirit makes the church into a body, organizing its various roles and parts, and a Christian's response must be humility with respect to his or her own role, joined with a love of Christ and of the unity of his whole body.

## 8. A Difference between and Two Continuities with *de Trinitate*

It would do to take a brief look at the connection of Augustine's Trinitarianism in a preached ascent narrative to his Trinitarianism in a more academic work like *de Trinitate*. In *Trin.* Augustine famously considers very subtle traces of the Trinity in human nature. The word *vestigium* for sign or trace is used in *Trin.* 6.10.12 (CCL 50, p. 242, Mountain and Glorie) and 11.1.1 (Augustinus 1968a, CCL 50, p. 333, Mountain and Glorie), and *indicium* for indication or evidence in *Trin.* 15.2.3 (Augustinus 1968b, CCL 50A, p. 462, Mountain and Glorie). For example, he considers the unity of the threefold activities of memory, intellect, and will as a trace of the Trinity. Likewise, the interplay of one loving, the thing loved, and love itself is a trace of the Trinity. How is this theology expressed in the sermons on the Psalms of Ascent? I make three observations on the subject, hopefully as prolegomena to future research.

First, it is not immediately obvious—at least not to me—how these traces or indications are explained to Augustine's flock in these sermons. It seems to me, therefore, that there is some degree of discontinuity, or at least a degree of difference, between *Trin.* and this sermon series.

Second, one notable continuity between *Trin.* and our sermon series is Augustine's description of the Son and the Holy Spirit in terms, respectively, of wisdom and love. In *Trin.* 15.17.29 Augustine explains that God's very substance is love, yet the Holy Spirit, in particular, is called love (*caritas*) (Augustinus 1968b, CCL 50A, p. 504, Mountain and Glorie). In the same way, God's wisdom is one with his substance, yet the Son, in particular, is called wisdom (*sapientia*). We saw above how in these sermons Augustine associates the Holy Spirit in particular with love. He identifies Christ the Son as the wisdom of God in *En.* 122, 5 (Augustinus 2001, CSEL 95/3, pp. 114–16, Gori). Again, in *En.* 126, 13 wisdom speaks in the gates of the city (drawing from Proverbs 8), and Christ is also a preacher in the gate (Augustinus 2001, CSEL 95/3, pp. 204–6, Gori).

Third, another continuity is in Augustine's approach to reason and authority when it comes to Trinitarian theology. For all Christians, there is the requirement of *En.* 130, 9: accept the orthodox doctrine of the Trinity. In *Trin.* Augustine says, "For among the faithful this ought not to have questioning"—*nam hoc inter fideles non debet habere quaestionem* (Trin. 15.6.9; Augustinus 1968b, CCL 50A, p. 472, Mountain and Glorie). But it is desirable to add faith to understanding. Thus Augustine has tried, in his lengthy books *de Trinitate*, to come by degrees (*gradatim*) to a better understanding of God, finding in created things some indications of their creator (15.2.3; Augustinus 1968b, CCL 50A, p. 462, Mountain and Glorie). We do not seek vainly, but at best we see, as Paul says, through a glass darkly (15.8.14–15.9.16; Augustinus 1968b, CCL 50A, pp. 479–83, Mountain and Glorie).

Perhaps there are other continuities between Augustine's Trinitarianism in the sermons on the Psalms of Ascent and in his other writings. I wish to call attention to this one in particular, for it explains our observed discontinuity. *de Trinitate* is a highly advanced effort to seek a better understanding of God. Augustine's *sermones ad plebem*, his sermons to the people, must start the faithful off on the milk of orthodoxy. It can then but recommend the ascent, without doing much to actually go through with it in the course of a few sermons to churchgoers who are at different levels—many just starting off. We should expect to see, in this context, little or no obvious development of the extremely subtle analogies from *Trin*. In the sermons on ascent in the Psalms, we see instead just what we should expect to see—an invitation. This is given explicitly in *En*. 61, 18. Here Augustine, laying out a few questions about Christ the Word of God, asks who will explain them. He then answers that *we* can try, and so he invites the faithful to climb up and see for themselves. *Ite cum Idithun, et videte*, he says—*Go with Iduthun, and see!* (*En*. 61, 18; Augustine 2000; Augustinus 1956b, CSL 39, p. 67, Müller; Augustinus 2020).

### 9. Conclusions

Augustine's sermons on the Psalms of Ascent demonstrate the power of his Trinitarian theology. Trinitarianism in these sermons is no idle dogma, but a transformative vision. It changes the whole nature of spiritual ascent. Ascent is not a matter of philosophical contemplation. It begins with a church professing the Trinity, ends with the delighted enjoyment of Jesus Christ, and all along involves the Holy Spirit healing the loves of the church. This transformed vision of spiritual ascent has practical effects on Christian living. It is, finally, a *preached* Trinitarianism. Instead of seeking to understand the Trinity as much as possible by studying traces of the Trinity in creation in the manner of *Trin.*, Augustine trains the church in the basics of orthodoxy and urges everyone to continue the climb towards a greater understanding of God.

From these sermons, we can learn some familiar lessons. Nicene orthodoxy has something to say about the church, something related to the critique of the Donatists (Ployd 2015). Augustine's Christology says something about how we are taken up into the life of God (Meconi 2013). Augustine connects the church singing the Psalms to the original biblical writers and readers and all the way up to the church in heaven, further connecting the teaching of the Psalms to the renewed lives of believers (McLarney 2014). Augustine's theology is a Nicene orthodoxy rather than a theology drawn from neo-Platonism (Ayres 2010). The emphasis on the different divine Persons in these sermons provides some modest support for Ayres' critique of (du Roy 1966). In du Roy's account of Augustine's Trinitarianism, "the distinctions between the persons—and their specific roles in the drama of salvation—are downplayed" (Ayres' words; Ayres 2010, p. 22). Augustine has treasured, since the pivotal moments in Milan when he read Plotinus and Paul, insights from the Platonists. Yet his preaching is drenched in Scripture, and he is committed to learning from the Bible how to think about God. He succeeds to some significant extent. As we see here, the ascent begins with a dogmatic commitment to the unity of all three of the Persons of the *Idipsum*, and the different divine Persons are involved in distinct ways in the church's salvation. There are no gradations of Godhood, but there are three Persons of the *Idipsum*.

We can learn these lessons and more. Above all, we can see how Platonic philosophy's ascent, while some of its components remain, is replaced with the life of the church growing in the love of God and neighbor. The project of climbing up the spiritual mountain has become the life of knowing God the *Idipsum* through living out the Gospel of Jesus Christ in the love of the Holy Spirit. The ultimate reality, the *Idipsum*, is the Holy Trinity, a doctrine foundational to an account of spiritual ascent which tells us how the church can ascend as the united body of Christ to the peaceful satisfaction of the enjoyment of God and of one another in God in the life of the heavenly Jerusalem.

**Funding:** This research received no external funding.

**Institutional Review Board Statement:** Not applicable.

**Informed Consent Statement:** Not applicable.

**Data Availability Statement:** No new data were created or analyzed in this study. Data sharing is not applicable to this article.

**Conflicts of Interest:** The author declares no conflicts of interest.

## Notes

1   I am grateful to some blind reviewers for *Religions* as well as to the Editors of this issue for their insightful comments and suggestions on an earlier draft of this article, and to John Martin for some linguistic assistance.

2   The conventional wisdom (presented, for example, by Chadwick 2001, p. 9) is that Augustine read little or, more likely, no Plato at all. It is possible that, as I was told by Carl Vaught, he did read Plato's *Timaeus*. For a more detailed introduction to his interactions with Greek philosophy, see (Fuhrer 2018, pp. 1687–93).

3   MacKenna's translation is based on the older critical text of Richard Volkmann. Dillon, as Editor of the Penguin edition, is attentive to errors stemming from this.

4   Fiedrowicz presents some scholarly views on the dating of these *enarrationes*, nearly all being in 406–7 or in 412 A.D. (Fiedrowicz 1997, p. 437). Müller presents some views on the dating, also noting how at the same time Augustine was preaching on the Gospel of John (*Io. eu. tr.*) with the common theme of the whole set of sermons being the critique of Donatism (Fiedrowicz and Müller 1996, p. 825). McLarney states that "The precise year in which they were delivered remains disputed but can be comfortably situated between 405 and 411" (McLarney 2014, p. 87). Ployd more specifically identifies the timeframe as a "seven-month period" stretching "from December of 406 to mid-summer of 407" (Ployd 2015, p. 2).

5   Where not otherwise indicated, English quotations from the *Enarrationes* are from Boulding's translations.

6   Boulding prefers "Song of Ascents", McLarney favors "song of steps", and the public-domain translation of portions of the *Enarrationes* at NewAdvent.org uses "song of Degrees".

7   Augustine shows in some of the *enarrationes*, as he does here, that he has been working hard to learn biblical Greek. I cannot say whether he was also able to consult Plotinus' Greek. McLarney (2014, pp. 47–52) overviews the scholarly discussion of Augustine's knowledge of biblical Greek.

8   I consider the general structure of the ascent more systematically but without a direct focus on the Trinity in Boone (2023, pp. 135–51). For a more general introduction to this series of sermons, see also (McLarney 2014). I consider the sermons on Iduthun in more detail in (Boone 2023, pp. 259–67). On Iduthun, see also (Grove 2021, chapter 3). I consider the multifaceted concept of identity in the *Enarrationes* in more detail in (Boone 2023, pp. 3–4 and pp. 284–85).

9   Modern translations, which I understand to be closer to the Hebrew, describe the infant in the comparison as *weaned* and contented, not contented with milk.

10  See (Ployd 2015) on how Augustine in these and related sermons touts an orthodox Christology against the Arians.

11  Orthodoxy is not merely accepting a dogma, but also participating in the life of the church. (See Ployd 2015, pp. 15–16).

12  Ayres prefers to leave it untranslated or to use the "double expression" of "*the selfsame, the identical*" (Ayres 2010, p. 202). The public-domain translation at NewAdvent.org simply uses "the same" or "the Same". Boersma considers the term and its translation in detail (Boersma 2018, pp. 251–52; see also McLarney 2014, p. 173).

13  Ayres on *En*. 121: "Throughout, 'God' refers to the Father who speaks through his Word. And yet because the Word or Son is equal to God he also has the name *idipsum*" (Ayres 2010, p. 205).

14  "This is all part of one movement, one ascent, one soteriology" (Ployd 2015, p. 55).

15  Acts of mercy are also the result of past ascent. (See (Grove 2021, pp. 94–98) on how ascent to the highest leads to an outward turn in the love of neighbor. See also (Ployd 2015, p. 105)).

16  Elaborating on how the love of the Holy Spirit works itself out in unity in the church, and connecting these insights to Augustine's critique of Donatism, is (Ployd 2015, chapter 3).

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
