# Peer review of "A Trinitarian Ascent: How Augustine’s Sermons on the Psalms of Ascent Transform the Ascent Tradition"

_religions, doi:10.3390/rel15050586_

Round 1

Reviewer 1 Report

Comments and Suggestions for Authors

This manuscript demonstrates a clear appreciation for the trinitarian dynamics of Augustine's theology of ascent. It is not clear, however, what the original contribution is. The author rightly references Ayres and Ployd, but it is unclear how this article goes beyond what hey have already exposited. While the author is definitely correct in their reading of Augustine, I do not believe that the article demonstrates sufficient originality for a peer-reviewed article in a top-tier journal. 

Author Response

Thank you for your comments!

I have tried to clarify what is the original contribution of my article.

I wonder if we have some philosophical differences on what counts as an original contribution. As far as I can tell, no one else has written on how Augustine’s Trinitarianism transforms the ascent genre from the Platonic tradition in philosophy. This is something important to ancient philosophy, something important to Augustine, an important effect of his Trinitarianism, and an interesting illustration of how he preaches to a popular audience largely unable to pursue the Trinitarian reflections of De Trinitate—as well as an interesting way of reading the Psalms!  If it’s important for these reasons, then it seems to me that some scholar should write about it.

Reviewer 2 Report

Comments and Suggestions for Authors

the author's study contains several positive elements. However, he neglected the essential context of the delivered homilies enarrationes in psalmos 119-133, which is a collection of 14, respectively 15 homilies, and this means that at the same time he also delivered part of the Interpretations of the Gospel of John, i.e. homilies 1-16 as well as the Commentary on the First John's letter: it is a cycle of parallel commented biblical texts, the themes of which are often repeated and complement each other. it is a period of years between December 406 and February 407: thus it is anti-Donatist texts; however, the author only gently reminds what is the essential interpretive context. the theme of unity and the journey towards it in the Church is rather pushed into the background. it would be interesting if the author put these ecclesiological anti-Donatist impulses into context with the Plotinian influence. Augustus borrowed the image of the soul's ascent from the material world to God, his highest good, from Plotinus, which is true, but the author does not sufficiently analyze the primary sources themselves. the text of the study suggests that secondary literature is more substantial than primary. points 2 and 3 are inserted separately without subsequent processing with the corpus enarrationes 119-133; therefore, they can be thrown out completely if they are not methodologically processed. the retelling of the early works from the point of view of the theme of pilgrimage (point 3) and ascent is more of the overview character that we find in the Augustinian handbooks. both points bring nothing new. however, without these points, the study will be significantly shortened and will require methodological revision.

Author Response

My thanks to you for these thoughtful comments!

I have, first of all, tried to foreground the context of the anti-Donatist emphasis of this period and the connections to these other writings on John’s contributions to the New Testament.  I have also clarified how the emphasis on unity connects to Plotinus, but how in my view it has much more to do with the Bible.

I have, secondly, interacted more with the primary sources.

Thirdly, I have more strongly emphasized the connections and differences between earlier Augustinian writings on ascent as well as between Augustine’s and the Platonic accounts of ascent.

Reviewer 3 Report

Comments and Suggestions for Authors

The article gives a thoughtful introduction to the theme of ascent in Augustine's En. Ps. The most intriguing passages of this article are a detailed analysis of some motifs in several of Augustine's sermons on Psalms. Thereby, the author detects both Trinitarian, Christological, and Pneumatological perspectives in Augustine's theory of ascent.

The article presents a fine depiction and summary of Augustine's Trinitarian interpretation of the spiritual ascent. This might be sufficient for the purpose of this special issue. However, the article does not clearly follow an own original thesis (although the author explicitly claims this [l. 299-304]).

Generally, the argumentation could be more pointed. Especially, the first chapters seem to be a bit lengthy and cursory (cf. a lack of references and discussions). Moreover, the article does not sufficiently discuss Augustine's relation to Platonic ascent-theories (Plato, Plotin) that their extensive depiction is justified. The convincing thesis that Augustine's Trinitarianism "changes the whole nature of spiritual ascent" (l. 542-543; cf. also l. 299-304) should be more accentuated.

Moreover, due to the presentation of Augustine's earlier works on ascent one should inquire the impact of the genre of sermons to Augustine's argumentation and proceeding in En. Ps. It could be stated more clearer which En. Ps. are exactly the sermons on the Psalms of Ascent, i.e. for what reasons the author choose which of Augustine's sermons.

Author Response

Thank you for your comments!

I have tried to clarify what is the original contribution of my article.  I wonder if we have some philosophical differences on what counts as an original contribution. As far as I can tell, no one else has written on how Augustine’s Trinitarianism transforms the ascent genre from the Platonic philosophers. This is something important to ancient philosophy, something important to Augustine, an important effect of his Trinitarianism, and an interesting illustration of how he preaches to a popular audience largely unable to pursue the Trinitarian reflections of De Trinitate—as well as an interesting way of reading the Psalms!  If it’s important for those reasons, then it seems to me that some scholar should write about it.

I have also made some efforts to accentuate the distinction between Platonic and Augustinian ascent.  Whether it is to your satisfaction is not for me to say, but I think it has at any rate made my paper better, and I am grateful for the recommendation.

Without writing an entirely new paper, it was possible briefly to address the impact of sermon genre on Augustine’s proceeding in these sermons.  This is in a new section just before the article’s Conclusion.

Reviewer 4 Report

Comments and Suggestions for Authors

Author Response

My thanks for the thoughtful and useful comments!

To begin with, I have considered and taken nearly all the stylistic suggestions.  (A few exceptions, such as Plotinus as “a great interpreter of Plato’s writings, a great synthesizer of their various accounts and images”; here the second phrase is used appositionally to the first, and your suggestion, while a fine phrasing, would change my meaning somewhat. And the second paragraph of Section 3 summarizes the conclusions of Section 3, but as it functions as the thesis statement for that section I opt to keep it at the beginning.)

Without making my article unduly lengthy or changing its topic dramatically, I don’t see how I could go into a great deal of detail on the scholarship on ascent in Republic and Plotinus. However, I have delved into the scholarship more deeply than I did in the original version of the article.

Now we come to your recommendation that I consider how Augustinian Trinitarian theology—such as the analogies considered in De Trinitate—pertains to these sermons.  I am truly grateful for this advice, and I hope I have done justice to its insight.  However, the result may not be what you expect.  I confess that I just don’t see Augustine talking about those analogies in these sermons in any clear way.  But I do see some truly remarkable connections to Trin., one of which explains why we should probably not expect to see such analogies here.  I have considered this in a new section, right before the paper’s conclusion.

Finally, I have added a few words to the paper’s conclusion on the problem with Augustinian interpretation you mention.  I hope it is enough to clarify things.

Round 2

Reviewer 1 Report

Comments and Suggestions for Authors

The author has done good work clarifying the contribution that the article makes and situating it more securely within the scholarly discussion. I believe it is now in an appropriate state for publication. 

Author Response

Thank you for your comments and your time!

Reviewer 4 Report

Comments and Suggestions for Authors

The paper is a serious and constructive contribution to Augustinian studies and, in particular, the relationship of Augustine's theology and his sermons of ascent.

Author Response

(The authors gave the same response as above.)
